# Evaluation of Optimal Sites for the Insertion of Orthodontic Mini Implants at Mandibular Symphysis Region through Cone-Beam Computed Tomography

**DOI:** 10.3390/diagnostics12020285

**Published:** 2022-01-23

**Authors:** Shizhen Zhang, Xiaoyu Wei, Lufei Wang, Zhouqiang Wu, Lu Liu, Xinyu Yan, Wenli Lai, Hu Long

**Affiliations:** 1State Key Laboratory of Oral Diseases, National Clinical Research Center for Oral Diseases, Department of Orthodontics, West China Hospital of Stomatology, Sichuan University, Chengdu 610041, China; 2018224035134@stu.scu.edu.cn (S.Z.); weixy8@mail2.sysu.edu.cn (X.W.); 1123968181@163.com (Z.W.); luliu@stu.scu.edu.cn (L.L.); xinyuyan100@163.com (X.Y.); 2Department of Oral and Craniofacial Health Sciences, University of North Carolina Adams School of Dentistry, Chapel Hill, NC 27599, USA; lufei_wang@unc.edu; 3Department of Orthodontics, West China School of Stomatology, Sichuan University, Chengdu 610041, China

**Keywords:** orthodontic mini implants, CBCT, mandibular symphysis

## Abstract

This study aims to evaluate the overall bone thickness (OBT) and cortical bone thickness (CBT) of mandibular symphysis and to determine the optimal sites for the insertion of orthodontic mini implants. Cone-beam computed tomography (CBCT) images of 32 patients were included in this study. The sample was further categorized into three facial types: low-, average-, and high-angle. OBT and CBT were measured at the mandibular symphysis region. All measurements were performed at six different heights from the cementoenamel junction [CEJ] and at seven different angles to the occlusal plane. Analysis of variance (ANOVA) was used for statistical comparison and a *p* value less than 0.05 was considered statistically significant. Our results revealed that neither OBT nor CBT was influenced by age or sex, except for the observation that CBT was significantly greater in adults than in adolescents. OBT and CBT were significantly greater in low-angle cases than in average- and high-angle cases. Both OBT and CBT were significantly influenced by insertion locations, heights and angles, and their interactions. CBT and OBT were greatest at the location between two lower central incisors, and became greater with increases in insertion height and angle. Both recommended and optimal insertion sites were mapped. The mandibular symphysis region was suitable for the placement of orthodontic mini implants. The optimal insertion site was 6–10 mm apical to the CEJ between two lower central incisors, with an insertion angle being 0–60 degrees to the occlusal plane.

## 1. Introduction

Orthodontic mini implants have been gaining in popularity among orthodontists due to their simplification of orthodontic biomechanics [1]. Various anatomic sites are available for the insertion of mini implants, e.g., inter-radicular sites, palatal sites and infrazygomatic areas [2,3,4]. In particular, anterior regions are frequently used for the insertion of mini implants to intrude between incisors [5]., Mini implants are less frequently used at the mandibular anterior region compared to the maxillary anterior region due to limited inter-radicular space, especially among mandibular crowding patients [3]. However, mini implants at the mandibular anterior region are clinically useful for lower incisor intrusion, intermaxillary fixation and molar protraction [6].

Previous studies that focused on inter-radicular sites for mini implants at the mandibular region suggested that mandibular incisor regions were not feasible for the insertion of mini implants [3,7]. Fortunately, in contrast to inter-radicular regions, the quality and quantity of alveolar bone labial to mandibular incisor roots (mandibular symphysis region) are adequate to accommodate mini implants without the risk of root damage. This renders the mandibular symphysis region a promising alternative for the insertion of mini implants at the mandibular anterior region. However, to date, no study has investigated bone characteristics at the mandibular symphysis region for orthodontic mini implants. Thus, our study aims to measure the bone thickness at the mandibular symphysis region through CBCT, and to determine optimal sites and insertion angles for the insertion of mini implants.

## 2. Materials and Methods

In total, 32 systematically healthy patients from the West China Hospital of Stomatology, Sichuan University were enrolled and their CBCT images retrieved. The inclusion criteria consisted of patients with fully erupted incisors and no congenital or developmental craniofacial anomalies. The exclusion criteria included (1) missing teeth or any dental implant in the anterior mandibular arch; (2) blurred or unclear images; (3) medical history related to bone metabolism; (4) previous orthodontic treatment. The sample was further grouped by sex (18 males and 14 females), age (16 adolescents aged 11–16 and 16 adults aged 18–29), and facial type (10 low-angle [MP-FH ≤ 22°], 11 average-angle [22° < MP-FH < 29°] and 11 high-angle [MP-FH ≥ 29°]). Informed consent was obtained from patients or their parents for those under 18.

CBCT examinations were performed with a three-dimensional volume scanner (MCT-1, J Morita Mfg Corp, Kyoto, Kyoto-fu, Japan). The settings of the scanner were as follows: 85 kV, 5.0 mA; exposure time of 17.5 s; and voxel size 0.2 mm.

As displayed in Figure 1a, seven anatomical locations were measured, i.e., 42, 42–41 (between right lateral incisor and central incisor), 41, 41–31, 31, 31–32 and 32 (corresponding to D, C, B, A, B’, C’, D’ respectively). At each location, measurements were performed at different heights (2 mm, 4 mm, 6 mm, 8 mm, 10 mm, and 12 mm from CEJ) and different angles (0, 10, 20, 30, 40, 50, and 60 degrees to the occlusal plane) (Figure 1b).

Both overall bone thickness (OBT) and cortical bone thickness (CBT) were examined. OBT was defined as the distance between the labial and lingual edge of the bone or between the labial edge and the lamina dura when the dental root was met (Figure 1b). CBT was interpreted as the distance between the external and internal aspect of the labial cortex (Figure 1b). The measurements were conducted by using INFINITT PACS (INFINITT Healthcare Co., Ltd., Seoul, Korea).

After a two-week interval, twenty percent of the sample were randomly selected for repeated measurement by the same investigator to test the intra-observer reliability.

The intraclass correlation coefficient (ICC) test and paired *t*-test were used to analyze the intra-observer reliability. The comparison of the left and right sides was performed by using the paired *t*-test.

Analysis of variance (ANOVA) was applied to investigate the influence of the following variables on OBT and CBT, including insertion location, facial type, sex, age, insertion height and insertion angle. Further comparisons among different insertion locations or among different facial types were performed via the Tukey post-hoc test. Two-way ANOVA was used to compare OBT and CBT among different insertion heights and insertion angles.

All data were analyzed by SPSS 25.0 and GraphPad Prism 8.3.0, and a *p* value less than 0.05 was considered statistically significant.

## 3. Results

### 3.1. Intra-Observer Reliability

The paired *t*-test revealed that the data were similar between the repeated measurements (*p* > 0.05). Moreover, the ICC test showed that the intra-observer reliability was good (r = 0.92).

### 3.2. Differences between Left and Right Sides

As displayed in Figure 1a, a total of seven locations were examined. We compared both OBT and CBT between the left and right sides, and found no significant difference between the two sides (*p* > 0.05 for both OBT and CBT). Thus, we combined the data of the two sides and a total of four locations (location A, B, C, and D) were examined for further analysis (Figure 1c).

### 3.3. Comparisons of OBT and CBT among Different Facial Types

As displayed in Figure 2, OBT was significantly thicker in the low-angle cases than in average- and high-angle cases at the insertion height of 12 mm at location A (*p* < 0.05). Moreover, similar results were found for CBT at location A and B (both *p* < 0.05) (Figure 3).

### 3.4. The Influences of Sex and Age on Measurements

As shown in Table 1, for OBT, no significant difference was found between sexes or between the two age groups (both *p* > 0.05). CBT was significantly higher among adults than among adolescents (*p* = 0.001), but did not differ between sexes (*p* > 0.05).

### 3.5. Comparisons of OBT and CBT at Different Insertion Locations

Tukey’s post-hoc test found that OBT was significantly different among the four insertion locations (*p* < 0.001) (Figure 4a). OBT was greatest at location A, with the order of thickness being A > C > B > D.

Similar results were found for CBT, except that no difference was found between location A and location C. The order of thickness was A = C > B > D (*p* < 0.001, Figure 4b).

Moreover, a three-way ANOVA test found that both OBT and CBT were influenced by insertion location (*p* < 0.001), insertion height (*p* < 0.001), insertion angle (*p* < 0.001) and their interactions (*p* < 0.001). Specifically, as depicted in Figure 5, OBT was similar among the four insertion locations at the insertion height of 12 mm, while significantly different among them at insertion heights lower than 12 mm (*p* < 0.001). Moreover, except for the insertion height of 2 mm, OBT increased with an increase in the insertion angle for all insertion locations. Similar results were found for CBT (Figure 6).

### 3.6. Comparisons of OBT and CBT among Different Insertion Heights and Insertion Angles

A two-way ANOVA test revealed that, at each insertion location, both CBT and OBT were significantly influenced by different insertion heights, insertion angles and their interactions (all *p* < 0.001). As displayed in Figure 7a, OBT increased with an increase in insertion height and insertion angle (except for the height of 2 mm). Similar results were found for CBT (Figure 7b).

### 3.7. Recommended and Optimal Insertion Sites

As shown in Figure 7, recommended insertion sites were identified as the areas where OBT was greater than 5 mm and CBT was 1–2 mm. Then, both recommended and optimal insertion sites were mapped from both the frontal and sagittal views in Figure 8.

## 4. Discussion

Anterior orthodontic mini implants have been well-validated for the effective correction of deep bite [5]. However, due to limited inter-radicular space in the mandibular anterior region, especially among patients with anterior crowding, the clinical application of mini implants for the intrusion of mandibular incisors was impeded [3]. Fortunately, it is suggested that extra-alveolar mini implants that are inserted outside the dental roots may expand the clinical applications of mini implants [8]. Mandibular symphysis is a complex articulation formed by the fusion of the left and right halves of the mandible [9]. As fusion progresses, mandibular symphysis grows anteriorly and laterally, resulting in an adequate bony projection anterior to the incisor roots [10]. This renders mandibular symphysis an excellent candidate for extra-alveolar orthodontic mini implants placement that could be used for the intrusion of mandibular anterior teeth.

Our recent study (not yet published) found that the mandible is more susceptible to asymmetry compared to the cranial bone and maxillae. Thus, we tested whether bone thickness was similar between the left and right sides. Our results revealed that neither OBT nor CBT differed between the two sides. This suggests that the insertion techniques are similar for both left and right sides, yet practitioners’ laterality should be considered.

It has been well-documented that the thickness of the alveolar bone was significantly influenced by different facial types [11,12]. Consistently, we found that both OBT and CBT were significantly thicker in the low-angle cases than in average- and high-angle cases. This finding is in line with Hoang et al. [13] but disagrees with Sadek et al. [12], which could be attributed to the different insertion heights at which measurement took place. Specifically, Hoang et al. measured the OBT at the apical level while Sadek et al. measured CBT at heights of 4 mm and 7 mm, which were not apical enough. Thus, we suggest that OBT and CBT differed among different facial types only at the apical level.

Sex and age factors play important roles in the development of bone thickness, while a previous study revealed that bone thickness was influenced by both sex and age [4]. However, we found that OBT and CBT were similar between males and females, which could be attributed to the different regions that were investigated. Interestingly, although OBT was similar between adults and adolescents, CBT was significantly greater among adults than adolescents, which is consistent with the study by Cassetta et al. [14]. These findings suggest that overall bone thickness may grow in the cortex, but not in the mandibular symphysis region, among adolescents.

Our results revealed that both OBT and CBT were thickest at location A and thinnest at location D. As mentioned above, the mandibular symphysis is formed by the fusion of two halves of a mandible and grows laterally as fusion progresses [10]. This median-to-lateral growth pattern explains why both OBT and CBT were thickest at location A and thinnest at location D. Interestingly, we found that OBT was different among the four locations (A > C > B > D) below 10 mm, but similar beyond 12 mm, which could be attributed to the influence of dental roots. Specifically, location A and C were inter-dental areas, while the dental roots were in location B and D, rendering OBT greater at the former locations than at the latter. In contrast, dental roots were not in the way of mini implant insertion beyond 12 mm for all the four sites; thus, OBT was similar among the four locations beyond 12 mm. Likewise, similar results were found for CBT. CBT was lower at root areas (B and D) than at inter-dental areas (A and C). This could be attributed to the fact that alveolar bone was expanded and the cortex thinned while the teeth erupted, resulting in a thinner or even lack of a cortex around the dental roots, which is supported by the phenomenon that alveolar bone defects in labial to dental roots are highly prevalent among the general population [15,16].

In clinical practice, in cases of inadequate bone quality and/or quantity, it is prudent to place orthodontic mini implants more apically and with certain angulations [17,18]. For example, it is advised to place a mini implant with an angle of 60–70 degrees to the occlusal plane at the infrazygomatic region to avoid root damage [8]. Due to limited inter-radicular space of the mandibular anterior areas, it is very likely that root damage is encountered during the insertion of mini implants at this area [19]. Thus, to avoid potential root damage at this area, inserting mini implants more apically with angulations is recommended. This could be explained by the following two factors. Firstly, dental roots become smaller and inter-radicular space is larger at a more apical level. Secondly, alveolar bone buccal to dental roots becomes thicker with larger insertion angulation. This is supported by our results that both OBT and CBT increased with an increase in the insertion height and insertion angle. Thus, we suggest that orthodontic mini implants could be placed apically with certain angulation at the mandibular anterior area, in order to avoid root damage.

The stability of orthodontic mini implants is influenced by several factors [20,21,22]. In particular, insertion depth and cortical thickness are of vital importance [23,24]. A great body of evidence reveals that mini implants are stable with an adequate insertion depth and appropriate cortical thickness [25,26]. It is well-documented that mini implants are stable if the insertion depth is greater than 5 mm [27]. Interestingly, theoretically speaking, mini implants could be more stable with greater cortical thickness. However, cortical fracture may happen during the insertion of a mini implant if cortical thickness is greater than 2 mm, rendering a cortical thickness of 1–2 mm appropriate for the insertion of mini implants [28]. Thus, we mapped the yellow areas with the following requirements: OBT greater than 5 mm and CBT of 1–2 mm (Figure 8a). In the yellow area, there is a trade-off relationship between the insertion height and insertion angle. Specifically, the insertion angle should be larger if the insertion height is lower, e.g., insertion angulation was 45–60 degrees at a height of 4–6 mm, but 0–60 degrees at a height of 12 mm. However, soft tissue inflammation is very likely if the insertion is too apical. On the other hand, mini implant slippage is more likely if the insertion angle is too large. Thus, clinicians should choose a not-too-apical insertion site in the yellow area, where soft tissue inflammation and slippage of mini implants are less likely to happen.

As mentioned above, the two halves of mandible that fuse at the median plane form a bony projection, rendering location A the best insertion site in the mandibular symphysis region. As depicted in Figure 8a,b, we suggest that the optimal insertion site is 6–10 mm apical to the CEJ between the two central incisors, with an insertion angulation of 0–60 degrees.

One limitation of this study is that we did not take sagittal discrepancy into consideration, since the thickness of the mandibular symphysis may vary among patients with different skeletal bases. A previous study revealed that mandibular symphysis thickness did not differ between Class I and Class III normo-divergent patients, suggesting that sagittal discrepancy does not affect symphyseal morphology [29]. However, future studies are warranted to delve into this notion.

Another limitation of this study is a relatively small sample size. Actually, since each patient has many CBCT slices, the sample size for this study had been increased by many times. This could mean that results from previous studies with sample sizes of 13, 30 and 60 were adequate for generalization [4,29,30]. However, future studies with larger sample sizes are warranted among different ethnicities from different countries for better generalization potential.

## 5. Conclusions

Mandibular symphysis is suitable for the insertion of orthodontic mini implants, with the best insertion location being between two central incisors.A mapping of recommended and optimal insertion sites with recommended insertion heights and insertion angles is suggested.The optimal insertion site is 6–10 mm apical to the CEJ between two central incisors, with an insertion angle of 0–60°.

## Figures and Tables

**Figure 1 diagnostics-12-00285-f001:**
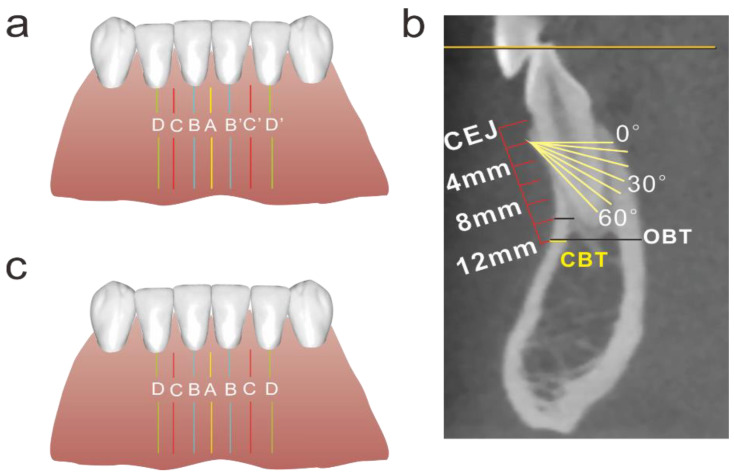
Schematic diagram of measuring method. (**a**) Seven insertion locations, A: the section bisecting the distance between two lower central incisors; B: labial to the middle of right central incisor; C: the section bisecting the distance between the right central and lateral incisors; D: labial to the middle of right lateral incisor; B’, C’, D’ represent the contralateral sites of B, C, D respectively. (**b**) Measurements of overall bone thickness (OBT) and cortical bone thickness (CBT) at different insertion heights and angles. (**c**) Four insertion locations after left- and right-side data were combined.

**Figure 2 diagnostics-12-00285-f002:**
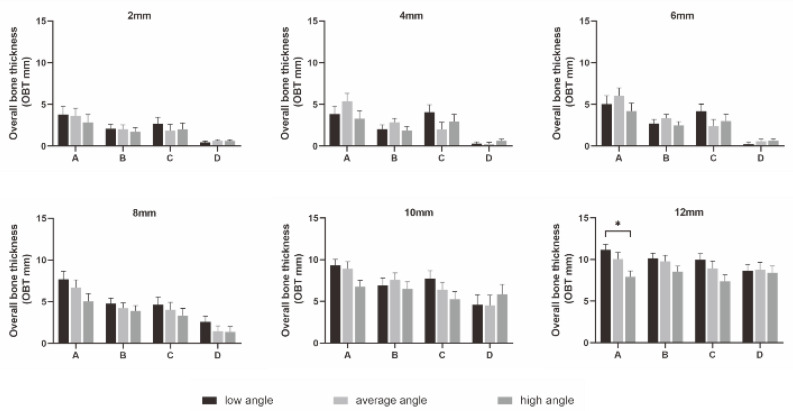
Comparisons of OBT among different facial types at each location. * *p* < 0.05.

**Figure 3 diagnostics-12-00285-f003:**
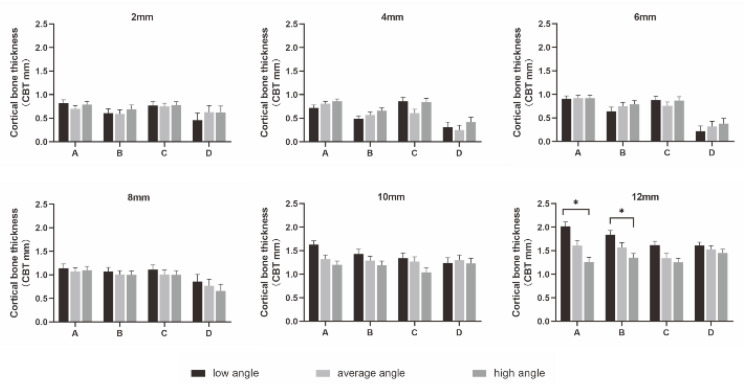
Comparisons of CBT among different facial types at each location. * *p* < 0.05.

**Figure 4 diagnostics-12-00285-f004:**
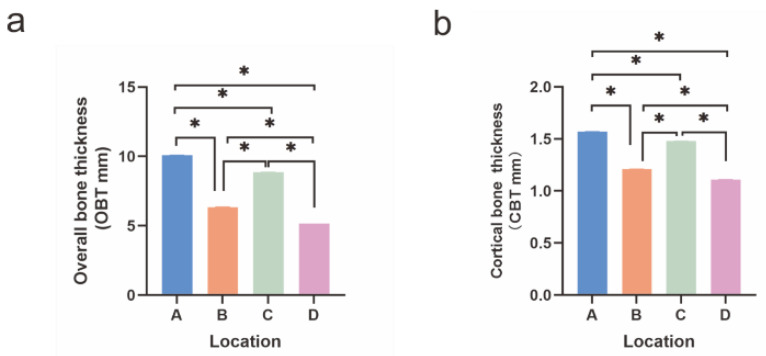
Comparisons of OBT and CBT among investigated locations. (**a**) Comparison of OBT among four insertion locations. * *p* < 0.05. (**b**) Comparison of CBT among four insertion locations. * *p* < 0.05.

**Figure 5 diagnostics-12-00285-f005:**
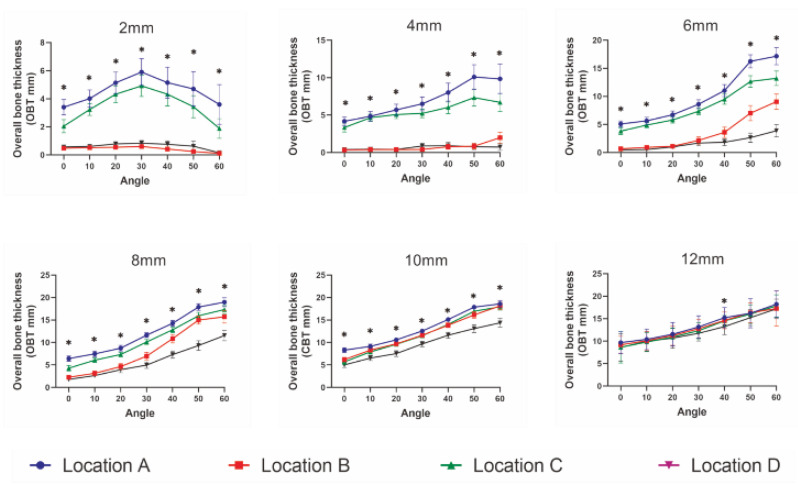
Comparison of the overall bone thickness (OBT) among four insertion locations at each insertion height. * *p* < 0.05, statistical significance among four sites.

**Figure 6 diagnostics-12-00285-f006:**
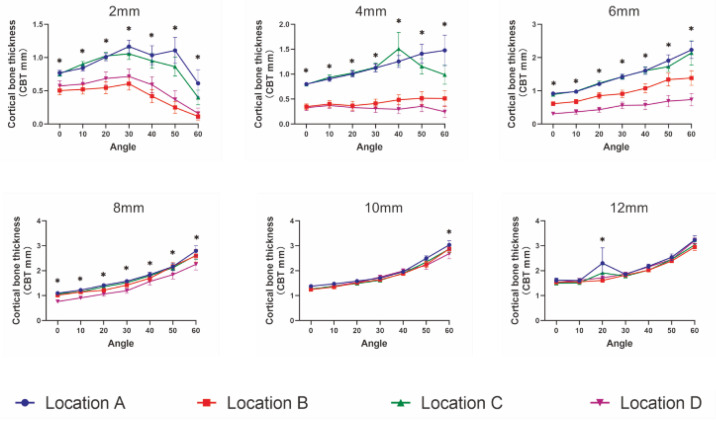
Comparison of the cortical bone thickness (CBT) among four insertion locations at each insertion height. * *p* < 0.05, statistical significance among four sites.

**Figure 7 diagnostics-12-00285-f007:**
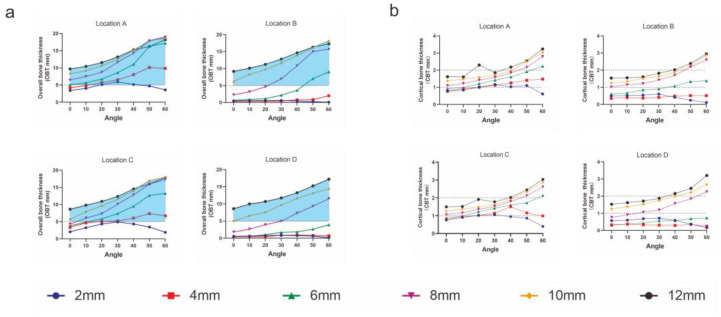
Comparison of OBT and CBT at different insertion heights and angles. (**a**) Overall bone thickness (OBT). The horizontal dotted lines are drawn at a thickness of 5 mm. The colored regions indicated the recommended height levels and insertion angles for mini-implant placement. (**b**) Cortical bone thickness (CBT). Thicknesses of 1 mm and 2 mm are marked with horizontal dotted lines.

**Figure 8 diagnostics-12-00285-f008:**
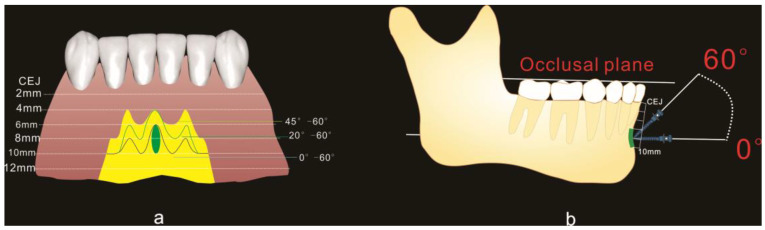
Recommended sites with appropriate insertion angles for mini implants at the mandibular symphysis region. (**a**) Frontal view. The yellow areas are feasible for mini-implant placement where OBT is greater than 5 mm and CBT is 1–2 mm. Different insertion angles are recommended for different subregions. The recommended insertion angle is 45–60 degrees for the yellow areas occlusal to the green line, 20–60 degrees for the yellow areas between the green and black line, and 0–60 degrees for the yellow areas apical to the black line. The oval green area is optimal for mini-implant placement, where OBT and CBT were the thickest, with an insertion height of less than 10 mm for the prevention of soft tissue irritation. (**b**) Lateral view of the optimal site and virtually placed mini implants in the optional range of insertion angles.

**Table 1 diagnostics-12-00285-t001:** Variance analysis of influence of gender and age on cortical bone thickness and overall bone thickness.

	Gender		*p*	Age		*p*
	Males	Females		Adolescents	Adults	
	Mean (mm) ± SD	Mean (mm) ± SD		Mean (mm) ± SD	Mean (mm) ± SD	
CBT	1.35 ± 1.06	1.33 ± 0.90	0.271	1.31 ± 0.89	1.38 ± 1.09	0.001 *
OBT	7.65 ± 6.93	7.57 ± 6.28	0.393	7.56 ± 6.76	7.67 ± 6.55	0.272

CBT, Cortical bone thickness; OBT, Overall bone thickness; * *p* < 0.05.

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
