# Peer review of "Evaluation of Optimal Sites for the Insertion of Orthodontic Mini Implants at Mandibular Symphysis Region through Cone-Beam Computed Tomography"

_diagnostics, 2022, doi:10.3390/diagnostics12020285_

Round 1
Reviewer 1 Report
Materials and Methods
Line 59. Sentence .. 16 adolescents (ages 18-29) and 16 adults (ages 11-16) -- swith the ages in the brackets
Literature
Reference 8 - missing volume and pages (pagination).
Reference 25 - missing volume and pages.
Author Response
Point 1: Line 59. Sentence .. 16 adolescents (ages 18-29) and 16 adults (ages 11-16) -- switch the ages in the brackets.
Response 1: Thank you very much for pointing out this mistake. We have corrected this mistake in the methods section accordingly. (Line 63, highlighted in yellow)
Point 2: Reference 8 - missing volume and pages (pagination). Reference 25 - missing volume and pages.
Response 2: Thank you very much for your thoughtful comment. We have added the volume and pages to reference 8 and reference 25. (Line 312 and 347 respectively, highlighted in yellow)

Reviewer 2 Report
The thickness of the vestibular cortical bone in patients with dentofacial deformities varies considerably with the dental compensations that occur in Angle class II and III. This occlusal characteristic is not specified as an exclusion criteria, perhaps it should be taken into account.
Perhaps the number of patients in the study could have been much larger, taking into account that they came from a University Hospital.

Author Response
Point 1: The thickness of the vestibular cortical bone in patients with dentofacial deformities varies considerably with the dental compensations that occur in Angle class II and III. This occlusal characteristic is not specified as an exclusion criterion, perhaps it should be taken into account.
Response 1: We thank you very much for this valuable and thoughtful comment. We totally agree with you on this point: symphyseal thickness may differ among patients with different sagittally skeletal discrepancy. We added this point in the limitation of this study, discussed it by citing one previous study, and called for future studies. (Line 260-265, highlighted in yellow)
Point 2: Perhaps the number of subjects in the study could have been much larger, taking into account that they came from a university hospital.
Response 2: Thank you very much for your critical suggestion. Actually, since CBCT images have many slices, thus one patient has abundant slices, making sample size being increased by many times. In fact, the sample sizes of previous studies were 13, 30 and 60 respectively, suggesting that the sample size of 32 in our present study was adequate. We admit that a small sample size is a limitation of our study and discuss this issue in the limitation section of this study. We called for future studies with larger sample sizes among different races from different countries for better generalization potential. (Line 266- 271, highlighted in yellow)

Reviewer 3 Report
Dear Authors
the article should be interesting but it has serious flaws to be published as in the current form.
I gave you some suggestions to improve the manuscript.
I encourage you to take advantage to address these considerations and resubmit the article.
Best regards
" Evaluation of optimal sites for the insertion of orthodontic mini-implants at mandibular symphysis region through cone-beam computed tomography”,:
ABSTRACT
- According to the article: “ Background: This study evaluates the overall bone thickness (OBT) and cortical bone thickness (CBT) of mandibular symphysis and to determine the optimal sites for the insertion of orthodontic mini-implants.” is this the aim of the present study?
- According to the article: “ Background: This study evaluates the overall bone thickness (OBT) and cortical bone thickness (CBT) of mandibular symphysis and to determine the optimal sites for the insertion of orthodontic mini-implants.” if the answer to the previous comment was YES, then the background subtitle should be changed to the aim subtitle because the content of the background subtitle describes the aim of the present study.
- The abstract does not present a specific objective
- According to the article: “Cone-beam computed tomography (CBCT) images of 32 patients (18 males, 14 females), including 16 adults and 16 adolescents were included in this study.” but the total sample of 32 people and also divided into 2 groups is adequate to obtain results that allow generalization in other populations?
- The abstract of the present study does not describe a global panorama of the content of the present study, it does not consider, for example, in material and methods the value of the level of significance (p <0.05) used in the statistical analysis of the results.
MATERIALS AND METHODS
- According to the article: “ The inclusion criteria consisted of patients with fully erupted incisors and no congenital or developmental craniofacial anomalies. The exclusion criteria included (1) missing teeth or dental implant in the anterior mandibular arch; (2) pathological lesions in the lower jaw” but the second inclusion criterion (non-developmental craniofacial anomalies) is the denial of the second exclusion criterion (pathological lesions in the lower jaw) and, as we know, an exclusion criterion should not be the denial of an inclusion criterion in a scientific investigation.
- According to the article: “ The sample was further grouped by sex (18 males and 14 females), age (16 adolescents [ages 18-29] and 16 adults [ages 11-16])…” but the age ranges between adolescents and adults should be [ages 11-16] and [ages 18-29] respectively because an adult is older than an adolescent.
- According to the article: “ facial type (10 low-angle [MP-FH≤22°], 11 average-angle [22°-29°] and 11 high-angle [≥29°])…” but with respect to the average-angle facial type, the mathematical expression <22 °-29 °> should be considered (in this way the values 22 ° and 29 ° are not included in this group).
CONCLUSIONS
- According to the article: “ INTRODUCTION…Thus, our study aimed to measure the bone thickness at mandibular symphysis region through CBCT, and to determine optimal sites for the insertion of mini-implants.” but the conclusion n ° 3 mentioned in this part of the article (“The optimal insertion site is 6-10 mm apical to CEJ between two central incisors with an insertion angle of 0o-60o.”) the angle of insertion has no relation with the proposed objectives.
Author Response
Point 1: According to the article: “Background: This study evaluates the overall bone thickness (OBT) and cortical bone thickness (CBT) of mandibular symphysis and to determine the optimal sites for the insertion of orthodontic mini-implants.” is this the aim of the present study? According to the article: “Background: This study evaluates the overall bone thickness (OBT) and cortical bone thickness (CBT) of mandibular symphysis and to determine the optimal sites for the insertion of orthodontic mini-implants.” if the answer to the previous comment was YES, then the background subtitle should be changed to the aim subtitle because the content of the background subtitle describes the aim of the present study. The abstract does not present a specific objective
Response 1: Thank you very much for your thoughtful and critical suggestion. This is the aim subtitle and we revised it accordingly. We changed the sentence in the abstract section to be “This study aims to evaluate the overall bone thickness (OBT) and cortical bone thickness (CBT) of mandibular symphysis and to determine the optimal sites for the insertion of orthodontic mini-implants”. (Line 17-19, highlighted in yellow)
Point 2: According to the article: “Cone-beam computed tomography (CBCT) images of 32 patients (18 males, 14 females), including 16 adults and 16 adolescents were included in this study.” but the total sample of 32 people and also divided into 2 groups is adequate to obtain results that allow generalization in other populations?
Response 2: Thank you very much for your critical suggestion. Actually, since CBCT images have many slices, thus one patient has abundant slices, making sample size being increased by many times. In fact, the sample sizes of previous studies were 13, 30 and 60, suggesting that the sample size of 32 in our present study was adequate. We admit that a small sample size is a limitation of our study and discuss this issue in the limitation section of this study. We called for future studies with larger sample sizes among different races from different countries for better generalization potential. (Line 266- 271, highlighted in yellow)
Point 3: The abstract of the present study does not describe a global panorama of the content of the present study, it does not consider, for example, in material and methods the value of the level of significance (p <0.05) used in the statistical analysis of the results.
Response 3: Thank you very much for this thoughtful and valuable comment. We are sorry for not including some contents in the abstract section of the previous manuscript due to word limits. We added these methods details in the abstract section in this revised manuscript. (Line 23 and 24, highlighted in yellow)
Point 4: According to the article: “The inclusion criteria consisted of patients with fully erupted incisors and no congenital or developmental craniofacial anomalies. The exclusion criteria included (1) missing teeth or dental implant in the anterior mandibular arch; (2) pathological lesions in the lower jaw” but the second inclusion criterion (non-developmental craniofacial anomalies) is the denial of the second exclusion criterion (pathological lesions in the lower jaw) and, as we know, an exclusion criterion should not be the denial of an inclusion criterion in a scientific investigation.
Response 4: Thank you very much for this meticulous and thoughtful suggestion. We totally agree with you and consider the second exclusion criteria to be unnecessary. We deleted the second exclusion criterium in the methods section. (Line 61 and 62)
Point 5: According to the article: “The sample was further grouped by sex (18 males and 14 females), age (16 adolescents [ages 18-29] and 16 adults [ages 11-16])…” but the age ranges between adolescents and adults should be [ages 11-16] and [ages 18-29] respectively because an adult is older than an adolescent.
Response 5: We thank you very much for pointing out this mistake. We are sorry for this mistake. Actually, we mistakenly put the adult age range for the adolescent group. We corrected this mistake in this revised manuscript. (Line 63, highlighted in yellow)
Point 6: According to the article: “facial type (10 low-angle [MP-FH≤22°], 11 average-angle [22°-29°] and 11 high-angle [≥29°])…” but with respect to the average-angle facial type, the mathematical expression <22 °-29 °> should be considered (in this way the values 22 ° and 29 ° are not included in this group).
Response 6: Thank you very much for this thoughtful suggestion. We have changed the mathematical expression for the average-angle facial type to be “22°<MP-FH<29°”. (Line 64, highlighted in yellow)
Point 7: According to the article: “INTRODUCTION…Thus, our study aimed to measure the bone thickness at mandibular symphysis region through CBCT, and to determine optimal sites for the insertion of mini-implants.” but the conclusion n ° 3 mentioned in this part of the article (“The optimal insertion site is 6-10 mm apical to CEJ between two central incisors with an insertion angle of 0o-60o.”) the angle of insertion has no relation with the proposed objectives.
Response 7: We thank you very much for this valuable and thoughtful suggestion. We added “insertion angles” in our study objectives (Line 54, highlighted in yellow).

Round 2
Reviewer 3 Report
Dear Authors
All the required modifications have been addressed. Now the article is suitable for publication.
Best regards